# Efficient screening of adsorbed receptors for *Salmonella* phage LP31 and identification of receptor-binding protein

Haojie Ge,[1,2,3] Ling Ye,[1,2] Yueyi Cai,[3] Huimin Guo,[1,2] Dan Gu,[1,2] Zhengzhong Xu,[1,2] Maozhi Hu,[1,2] Heather E. Allison,[3] Xin'an Jiao,[1,2] Xiang Chen[1,2]

**ABSTRACT** The adsorption process is the first step in the lifecycle of phages and plays a decisive role in the entire infection process. Identifying the adsorption mechanism of phages not only makes phage therapy more precise and efficient but also enables the exploration of other potential applications and modifications of phages. Phage LP31 can lyse multiple *Salmonella* serotypes, efficiently clearing biofilms formed by *Salmonella enterica* serovar Enteritidis (*S.* Enteritidis) and significantly reducing the concentration of *S.* Enteritidis in chicken feces. Therefore, LP31 has great potential for many practical applications. In this study, we established an efficient screening method for phage infection-related genes and identified a total of 10 genes related to the adsorption process of phage LP31. After the construction of strain C50041Δ*rfaL*$^{58–358}$, it was found that the knockout strain had a rough phenotype as an O-antigen-deficient strain. Adsorption rate and transmission electron microscopy experiments showed that the receptor for phage LP31 was the $O_9$ antigen of *S.* Enteritidis. Homology comparison and adsorption experiments confirmed that the tail fiber protein Lp35 of phage LP31 participated in the adsorption process as a receptor-binding protein.

**IMPORTANCE** A full understanding of the interaction between phages and their receptors can help with the development of phage-related products. Phages like LP31 with the tail fiber protein Lp35, or a closely related protein, have been reported to effectively recognize and infect multiple *Salmonella* serotypes. However, the role of these proteins in phage infection has not been previously described. In this study, we established an efficient screening method to detect phage adsorption to host receptors. We found that phage LP31 can utilize its tail fiber protein Lp35 to adsorb to the $O_9$ antigen of *S.* Enteritidis, initiating the infection process. This study provides a great model system for further studies of how a phage-encoded receptor-binding protein (RBP) interacts with its host's RBP binding target, and this new model offers opportunities for further theoretical and experimental studies to understand the infection mechanism of phages.

**KEYWORDS** phage, adsorption receptor, receptor-binding protein, *Salmonella*, $O_9$ antigen

Members of the *Salmonella* genus are Gram-negative, facultative, intracellular bacteria comprising more than 2,500 serotypes (1, 2). They can infect animals, including poultry and pigs, and may pose serious and life-threatening infections in humans (3, 4). *Salmonella* infections in humans are usually associated with the consumption of *Salmonella*-contaminated food or water or due to direct contact with infected animals (5). According to the data released by the European Centre for Disease Prevention and Control, approximately 52,702 cases of *Salmonella* infections were reported in 2020, accounting for 22.5% of all foodborne human diseases (6). *Salmonella enterica*

Address correspondence to Xin'an Jiao, jiao@yzu.edu.cn, or Xiang Chen, chenxiang@yzu.edu.cn.

The authors declare no conflict of interest.

See the funding table on p. 17.

serovar Enteritidis (*S*. Enteritidis) is the most common cause of invasive human gastroenteritis disease and non-typhoidal *Salmonella* infections (2). *S*. Enteritidis ranked as the most common of 15 frequently identified serovars from human samples between 2001 and 2007 in laboratories in 37 countries (7). Between 2016 and 2018, 346 *Salmonella* strains were isolated from three large-scale chicken farms across different provinces of China. Of these 346 strains, 329 strains were *S*. Enteritidis, accounting for 95.09% of the *Salmonella* isolates (8). Therefore, prevention and control of *S*. Enteritidis infection is of great significance for the development of the poultry breeding industry and human health.

The overuse of antimicrobial drugs in agriculture and medicine in recent decades has led to the emergence of multi-drug-resistant (MDR) *Salmonella* (9–11). *S*. Enteritidis is the most serious antimicrobial-resistant serotype found in healthy chickens in central China, with high resistance rates to antibiotics such as colistin, meropenem, and ciprofloxacin (12). Sun et al. (13) isolated 525 *S*. Pullorum strains from China that were resistant to at least one antibiotic, among which 280 strains (42.9%) were resistant to three or more antibiotics. The emergence of antibiotic resistance reduces the efficacy of antibiotics and increases the incidence and mortality of *Salmonella* infections in animals and humans (6). Due to the lack of new antibiotics on the market, the emergence of MDR bacteria makes it increasingly difficult to prevent and control *Salmonella* infections. In addition, China, the United States, and the European Union have banned the use of multiple antibiotics in animal husbandry, making biosecurity more difficult and expensive to maintain at low risk levels on farms (14, 15). Therefore, there is an urgent need for safe, effective, and low-cost means to control *Salmonella* infections in animal husbandry and the associated processes, including food, transportation, and storage.

Bacteriophages, or phages, are viruses that can specifically infect and lyse bacteria, and these viruses are widely distributed in nature (3, 16). In fact, most phages are very specialized in their interactions with bacteria, having evolved to recognize differences in bacterial species or even strains. Phages have been used as an alternative and ecologically friendly biological control agent for preventing and controlling MDR bacteria (17–19). However, the use of phages as therapeutic agents still faces challenges such as having an extremely narrow host range and being subject to interference by phage-resistance mechanisms in bacteria (20, 21). The limited host range is, in part, determined by the specificity of the host recognition system of phages. The phage receptor-binding proteins (RBPs) specifically recognize receptors on the surface of bacteria. Phages adsorb to the bacterial surface using these RBPs, which then enable infection (22, 23). However, recognizing and being able to adsorb to the bacterial cell surface does not guarantee that a phage will be able to infect a bacterial cell. Bacteria have multiple resistance mechanisms against phage infections, e.g., blocking adsorption and blocking injection of phage DNA into the host cell, expressing restriction and modification systems, operating abortive infection systems, and expressing CRISPR-Cas systems (24–27). Among these phage defense mechanisms, the ability to block phage adsorption often occurs when the cell surface receptor of the bacterial host mutates, disabling the phages from recognizing the host (28). Therefore, understanding how phages recognize their specific hosts can impact several areas of phage utility: (i) detailed knowledge of phage recognition of the host receptor can enable scientists to modify phages with the specific aim of expanding their host range; (ii) knowledge of phage (RBP) binding to the host's receptor can be utilized to optimize the design of phages or phage tail components for rapid identification of bacteria; and (iii) knowledge of phage RPB interactions with host receptor molecules can be utilized to optimize the design of phages to treat bacterial infections or other problems.

Phage LP31 is a lytic phage. Our previous research has shown that LP31 is able to lyse multiple serotypes of *Salmonella* and LP31 can also efficiently eliminate biofilms formed by *S*. Enteritidis and *S*. Pullorum. From our preliminary research, it was found that phage LP31 can significantly reduce the concentration of *Salmonella* both on metal surfaces and in chicken intestinal feces (29). Therefore, phage LP31 may have a significant role

to play in the control of *Salmonella*-mediated foodborne disease. In order to maximize our ability to efficiently utilize phage LP31 and other related phages, it is necessary to identify and characterize both the RBP of phage LP31 and the surface receptor it binds to on *Salmonella*. In this study, an efficient method for screening phage receptors is established, and multiple mutants from *S*. Enteritidis C50041 libraries are screened for phage resistance. The RBP of the phage (which in tailed phages is associated with the tail) was identified through receptor neutralization experiments, adsorption rate measurement assays, and transmission electron microscopy (TEM). This study provides some of the basic information necessary to understand *Salmonella* phage LP31 before it can be used in future targeted therapeutic or other biocontrol strategies. In addition, an efficient new way to identify phage receptors in bacteria is provided.

## RESULTS

### Screening for the LP31 phage-resistant strains

A random insertion transposon mutant library of *S*. Enteritidis C50041 was constructed, and the library was mixed with the phage LP31. The phage was allowed to infect and kill susceptible cells. A significant reduction in the number of viable cells was observed (Fig. 1A). Colonies (200, now known as putative LP31-resistant mutants) that survived this phage infection were picked. Drop spot assays identified that phage LP31 could not form clear or translucent zones on 19 of 200 putative LP31-resistant mutant lawns (Fig. 1B). Co-culture experiments *in vitro* revealed that while phage LP31 was able to effectively inhibit the growth of wild-type C50041 within 7 h, it was unable to significantly inhibit the growth of the 19 mutant strains (Fig. 1C). Therefore, these 19 strains became known as true phage-resistant mutants, or simply LP31-resistant mutants.

### Identification of the lipopolysaccharide synthesis genes involved in phage adsorption

Based on the results from PCR amplification and sequencing, we found that the transposon insertion sites in the LP31-resistant mutants mainly existed in 10 lipopolysaccharide (LPS) synthesis-related genes (Fig. 2A and B; Table 1). To verify the type of receptor phage LP31 recognizes on the bacterial surface, the adsorption capacity of phage LP31 before and after treatment with *S*. Enteritidis C50041 with sodium periodate or proteinase K, respectively, was measured. It was found that the adsorption of the phage to the host bacteria after proteinase K treatment did not change significantly, but the adsorption to the bacteria after sodium periodate treatment decreased significantly (Fig. 2C). The results show that after digestion of surface-exposed protein, the adsorption potential of phage LP31 remained unchanged, but after the polysaccharide was damaged, the adsorption potential of LP31 was significantly decreased, indicating that the polysaccharide is necessary for phage LP31 adsorption. In addition, adsorption experiments were used to measure the impact on adsorption of each transposon insertion from the 19 LP31-resistant mutants; all were shown to possess a significantly reduced ability to support phage LP31 adsorption (Fig. 2D). Therefore, we speculated that LPS may be the receptor for phage LP31.

### The deletion strain C50041Δ*rfaL*$^{58-358}$ shows phage resistance and a rough phenotype with O-antigen deficiency

An isogenic mutant of C50041 deficient in the production of the *rfaL* gene product was created, C50041Δ*rfaL*$^{58-358}$ (Fig. S1), and the mutation was complemented by enabling the expression of *rfaL* from an inducible plasmid, C50041Δ*rfaL*$^{58-358}$-p*rfaL* (Fig. S2), to verify whether LPS is the essential receptor for phage LP31. Phage LP31 was able to form clear and translucent plaques on the wild-type and complementation strains but not on the deletion strain (Fig. 3A). *In vitro* cultivation showed (Fig. 3B) that the growth curves of the wild-type, knockout, and complementation strains were roughly the same. After adding LP31 to the culture medium, the growth of the wild-type and complementation strains

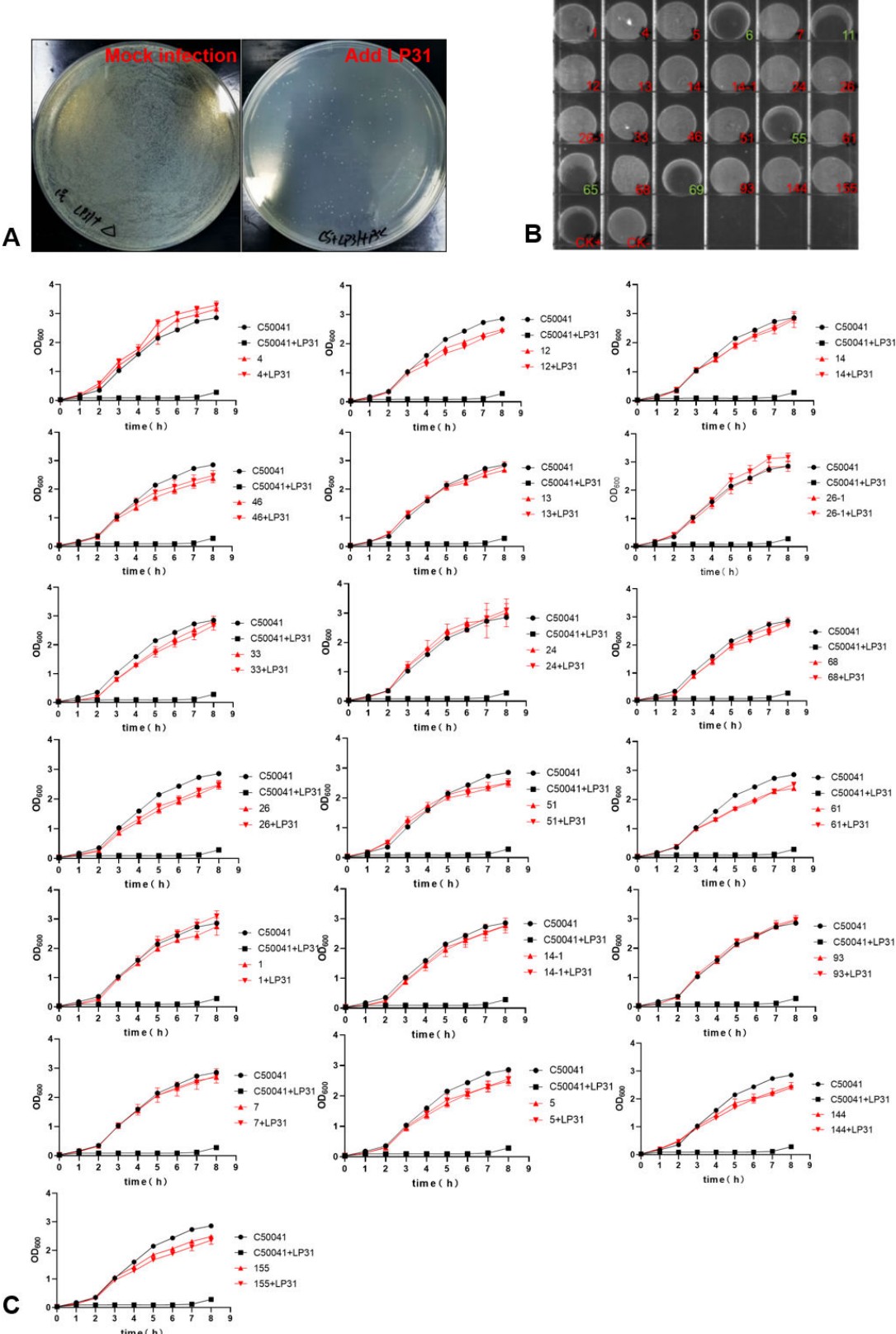

**FIG 1** Screening of phage-resistant strains. (A) Comparison before and after killing sensitive bacteria with phage. (B) Screening of the putative phage-resistant mutants by drop spot assay. The red numbers are true phage-resistant mutants. The green numbers are sensitive strains. (C) Growth curves of wild-type C50041 and LP31-resistant mutants with and without phage LP31, cultured *in vitro*.

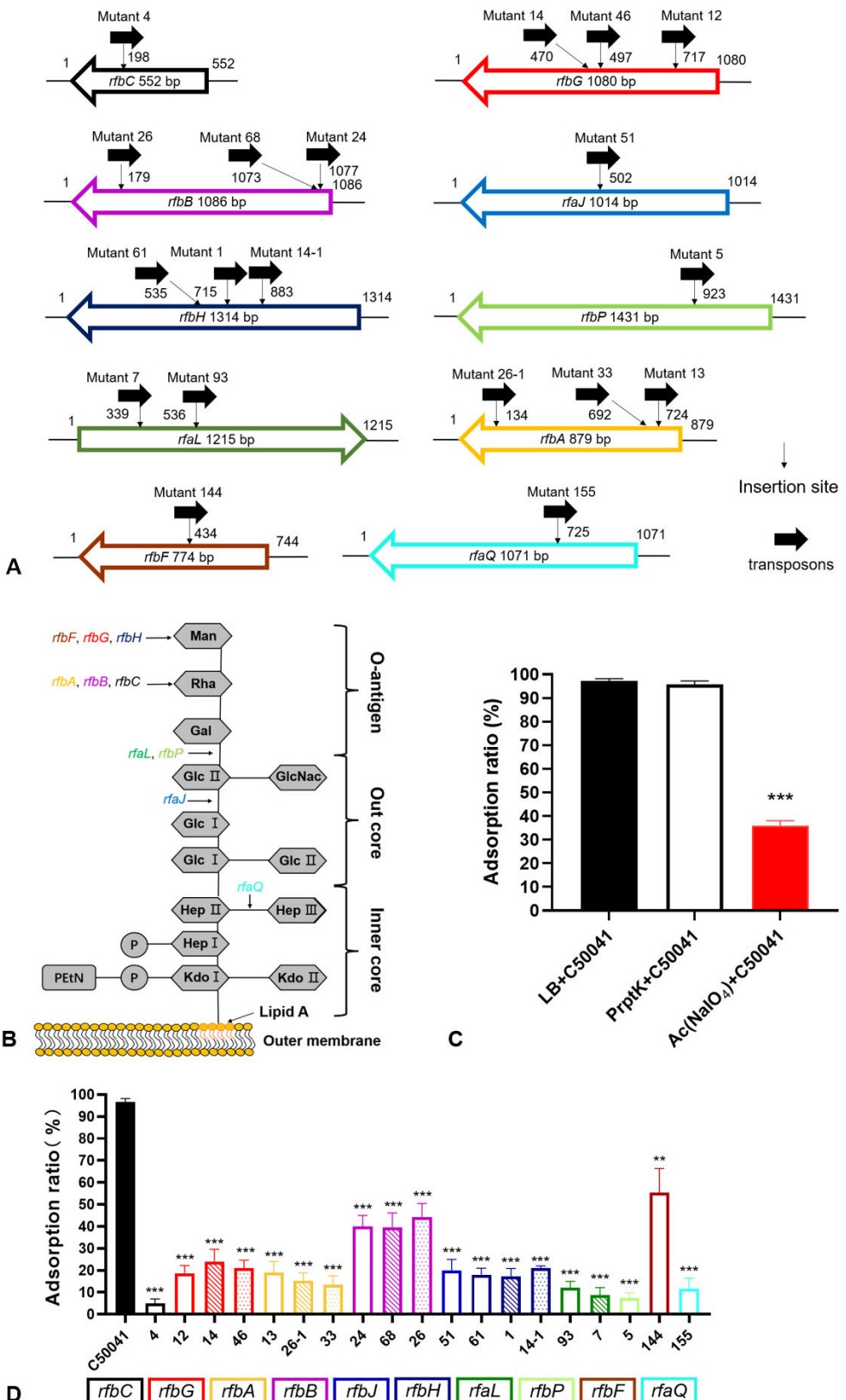

FIG 2   Screening of genes related to the phage adsorption process. (A) Analysis of transposon insertion sites in phage
LP31-insensitive strains. (B) Schematic structure of LPS, showing that different gene products (color-coded to match panel

**FIG 2 (Continued)**

A) required for LPS synthesis affect the adsorption of phage LP31. Abbreviations are as follows: Glc, glucose; Rha, rhamnose; GlcNAc, N-acetylglucosamine; Man, mannose; Hep, heptose; Gal, galactose; Kdo, 2-keto-3-deoxyoctulosonic acid; P, phosphate; PEtN, phosphoethanolamine. The structure of LPS is that of *S.* Typhimurium, modified from references (30, 31). (C) Adsorption properties of phage LP31 on *S.* Enteritidis C50041 or C50041 treated with proteinase K or Ac (NaIO$_4$). (D) Comparative LP31 adsorption measurements on different LP31-resistant mutants. The X-axis is labeled with the various Tn mutants, and they have been color-coded to match the gene identities defined in panel A. **, $P < 0.01$; ***, $P < 0.001$.

was almost completely inhibited within 7 h, while the growth of the knockout strain was not affected significantly, indicating that the knockout strain C50041$\Delta rfaL^{58-358}$ is resistant to phage LP31.

To verify whether the deletion of the *rfaL* gene affects the integrity of LPS on the surface of bacteria, the agglutination and auto-aggregation characteristics of the wild-type, deletion, and complemented strains were tested. The results showed that an O$_9$ mAB (monoclonal antibody against O-antigens) was able to support agglutination of the wild-type and complemented strains but not the *rfaL* deletion mutant. However, acriflavine, which supports the agglutination of rough (LPS mutant) strains, did support the agglutination of the *rfaL* deletion mutants but not the wild-type or complemented deletion mutant strains (Fig. 3C). Additionally, there was no auto-aggregation when the wild-type and complemented strains were incubated statically, while the deletion strain exhibited significant auto-aggregation with an aggregation rate of up to 78% (Fig. 3D). These results suggest that the deletion strain C50041$\Delta rfaL^{58-358}$ exhibits a rough phenotype with O-antigen deficiency.

## The O$_9$ antigen of LPS as an adsorption receptor for phage LP31

To determine whether the O-antigen on the *S.* Enteritidis LPS is the adsorption receptor for phage LP31, adsorption assays and subsequent TEM imaging were performed to observe the effect of the O-antigen of the *S.* Enteritidis LPS on phage adsorption. After incubation with smooth LPS (Sigma-Aldrich; purified from *S.* Enteritidis, 100 mg/mL), the adsorption capacity of phage LP31 to *S.* Enteritidis C50041 was significantly reduced (Fig. 4A), and no phage particles were observed on the surface of C50041 by TEM (Fig. 4C), indicating that phage LP31 bound to the soluble *S.* Enteritidis LPS, which affected the numbers of phage LP31 that were left to adsorb to the whole bacterial cells. In addition, the adsorption of phage LP31 to the *S.* Typhimurium mutant D6$\Delta rfbN^{83-188}$, *S.* Enteritidis C50041$\Delta rfbG$, and C50041$\Delta rfaL^{58-358}$ (all possessing an O-antigen deficiency) was significantly lower than that of wild-type strains (Fig. 4B). TEM images following the adsorption assay showed that a large number of phage particles were adsorbed on the surface of the wild-type *S.* Enteritidis C50041, but no phage particles were observed on the surface or even nearby the surface of C50041$\Delta rfaL^{58-358}$ (Fig. 4C). These results indicate that the adsorption process of phage LP31 requires the O$_9$ antigen of *S.* Enteritidis LPS to participate.

**TABLE 1** Transposon insertion information for phage-insensitive mutants

| Strain number | Gene | Protein | Pathway |
|---|---|---|---|
| 4 | *rfbC* | dTDP-4-dehydrorhamnose 3,5-epimerase | O-antigen synthesis |
| 12, 14, 46 | *rfbG* | CDP-glucose 4,6-dehydratase | O-antigen synthesis |
| 13, 26-1, 33 | *rfbA* | Glucose-1-phosphate thymidylyltransferase | O-antigen synthesis |
| 24, 68, 26 | *rfbB* | dTDP-glucose 4,6-dehydratase | O-antigen synthesis |
| 51 | *rfaJ* | Lipopolysaccharide 1,2-glucosyltransferase | LPS synthesis |
| 61, 1, 14-1 | *rfbH* | CDP-6-deoxy-D-xylo-4-hexulose-3-dehydrase | O-antigen synthesis |
| 93, 7 | *rfaL* | O-antigen ligase | LPS synthesis |
| 5 | *rfbP* | Undecaprenyl-phosphate galactose phosphotransferase | O-antigen synthesis |
| 144 | *rfbF* | Glucose-1-phosphate cytidylyltransferase | O-antigen synthesis |
| 155 | *rfaQ* | Lipopolysaccharide core heptosyltransferase | LPS synthesis |

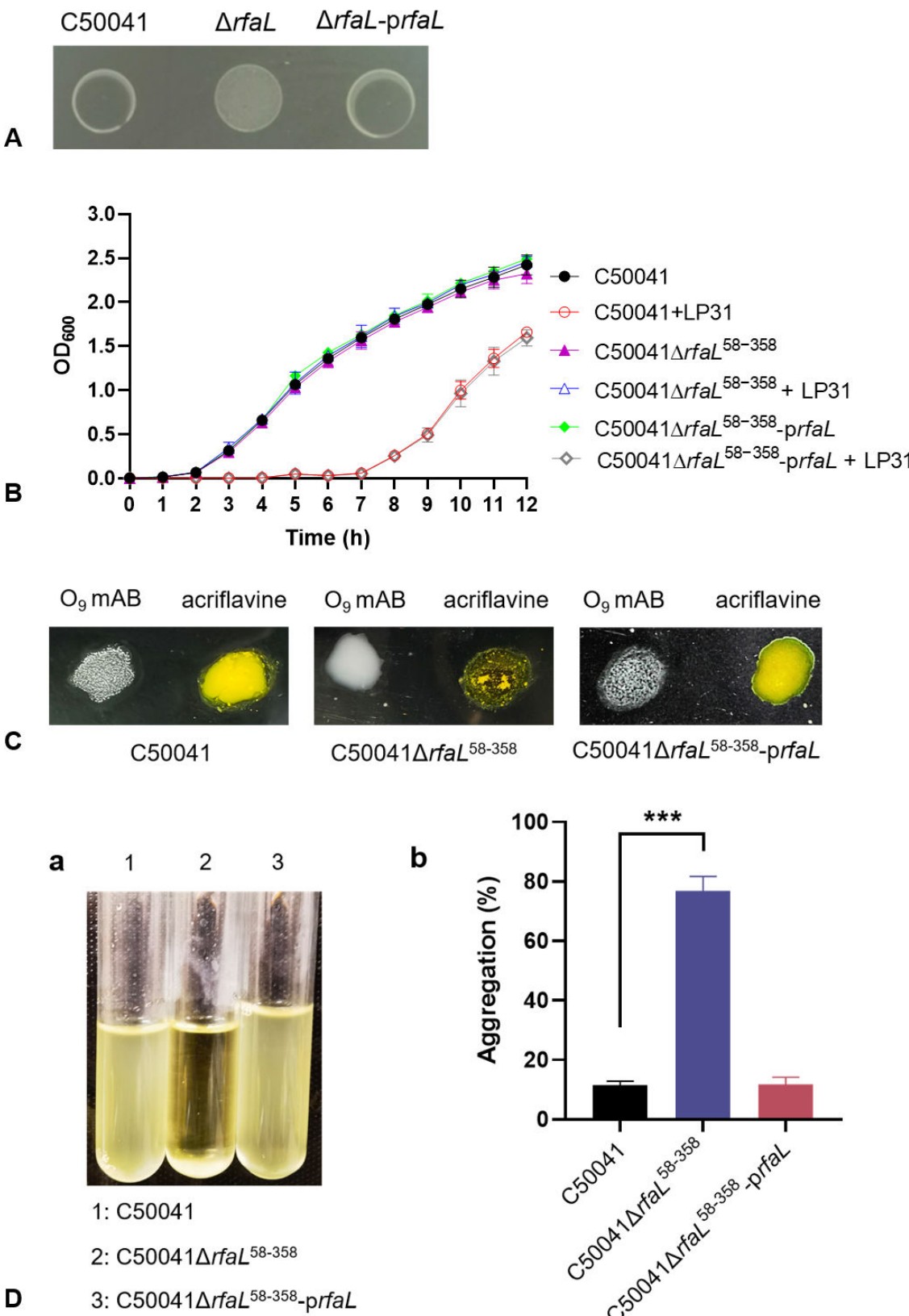

**FIG 3** Characterization of the *rfaL* gene deletion mutant (C50041Δ*rfaL*⁵⁸⁻³⁵⁸) and complemented strain (C50041Δ*rfaL*⁵⁸⁻³⁵⁸-p*rfaL*). (A) The ability of phage LP31 to form clear spots on phage drop spot assays on the wild-type C50041 strain, its *rfaL* gene deletion mutant variant, and the complemented strain. (B) Growth curves of the wild-type, *rfaL* deletion mutant, and the complemented mutant cultured *in vitro* with or without phage LP31. (C) Agglutination phenotypes of the

**FIG 3** (Continued)

wild-type, *rfaL* deletion mutant, and complemented mutant. *S.* Enteritidis with O-antigen supports agglutination by an $O_9$ mAB (monoclonal antibody against O-antigens) but does not agglutinate with acriflavine. The strain with the O-antigen synthesis defect does not agglutinate with $O_9$ mAB but does agglutinate in the presence of acriflavine. (D) The aggregation of different strains: (a) aggregation of bacteria during cultivation; (b) percentage of aggregation of different strains (***, $P < 0.001$).

## Identification of the receptor-binding protein Lp35

To identify the RBP of phage LP31, all potential tail-associated proteins encoded by phage LP31 were bioinformatically predicted (27). A total of five proteins identified as phage tail or tail-associated proteins (Lp24, Lp34, and Lp35) were cloned with fused histidine tags and purified (Fig. 5A; Fig. S3, Lp24 and Lp34 data not shown). The products of two genes, Lp30 and Lp23, were predicted to be a tape measure protein and the main tail protein, respectively, and were not chosen for further study. Adsorption assays showed that only the purified recombinant Lp35 protein was able to significantly decrease the adsorption of phage LP31 to *S.* Enteritidis C50041 (Lp35 + $H_2O$) (Fig. 5B). However, no significant change in the adsorption of phage LP31 was detected using either purified Lp24 or Lp34 (data not shown). Bioinformatic analysis of protein Lp35 (encoded by the *lp35* gene) indicated that Lp35 shared the greatest level of homology to a tail fiber protein of phage vB_SenS_SE1 (Fig. 5C and D) (32). Subsequently, after incubating with LPS and Lp35 and then blocking C50041, we found that the adsorption of phage LP31 to C50041 was significantly higher than that to C50041 only blocked by Lp35 (Lp35 + $H_2O$) (Fig. 5B). TEM observation also showed that there was very limited phage binding to the surface of C50041 that was first blocked by Lp35 (Lp35 + $H_2O$), but that phage LP31 could adsorb to C50041 that was first blocked by LP35 incubated with LPS (Fig. 4C). All of these results indicate that the tail fiber protein Lp35 of phage LP31 participates in the adsorption process as an RBP.

## DISCUSSION

The life cycle of lytic phages generally consists of several stages: host adsorption, injection, DNA replication, gene transcription, protein translation, phage assembly, and finally lysis of host cells to release progeny phages (24). Adsorption of the phage to the bacterial surface is the first step in phage infection. The phage binds to a specific receptor on the surface of the host, allowing it to recognize a suitable host in a mixed population of bacteria. Meanwhile, bacteria can develop resistance to phage infection by mutating the specific receptors on their cell surfaces, thereby preventing phage adsorption (33). Therefore, specific receptors are crucial for the phage infection process. Currently, the development of phage therapy and phage display technology largely relies on the phage adsorption mechanism (30, 34, 35). However, the traditional method of co-cultivating phages and hosts to induce host mutations and phage resistance and identify genes related to phage adsorption is time-consuming and can only identify a limited number of target genes. In addition, predicting phage adsorption receptors through bioinformatic methods lacks experimental data support, making its accuracy and reliability for characterizing phage-host interactions for therapeutic strategies less certain, which is an important step in realizing the full potential of phage therapy in clinical trials and use (36). Previously, we screened for phage resistance strains from a mutant library using a drop method, which was laborious and time-consuming (screening 3 phage resistance strains from 5,000 mutant strains) (28). Therefore, in this study, we optimized the previous screening method (added process 2 in Fig. 6 to the original method) and established an efficient screening method that only took 3 days, significantly improving efficiency (Fig. 6). Using this method, we screened 19 phage LP31-resistant mutants from 200 putative *S.* Enteritidis C50041 mutants (Fig. 1B and C). The mutated genes of these strains were all related to LPS synthesis (Fig. 2A and B and Table 1) (37). Common phage adsorption receptors in Gram-negative bacteria include LPS, outer membrane proteins, flagella, pili, and capsules (31). We deduced that the receptor

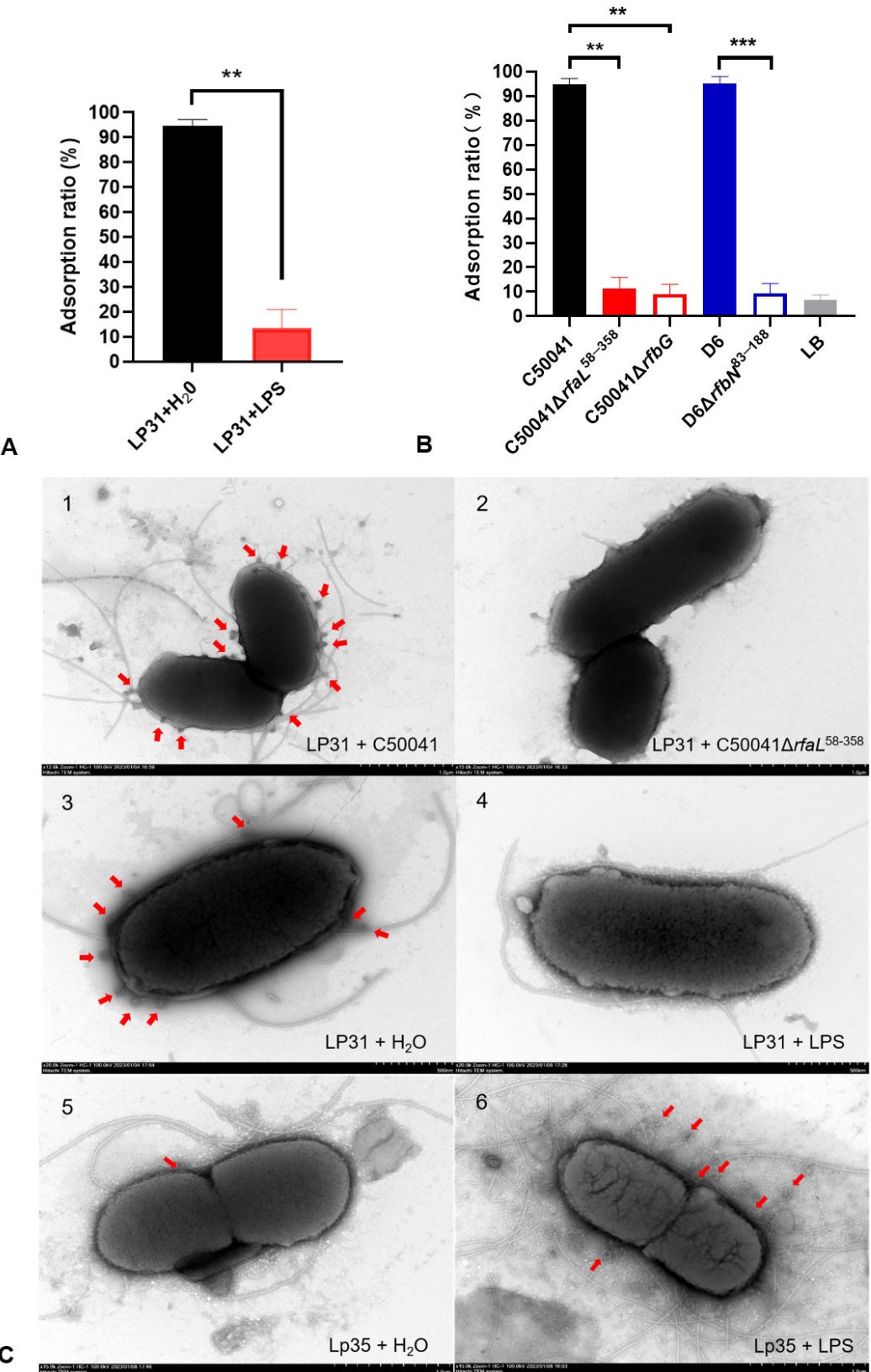

**FIG 4** Characterization of phage LP31 adsorption. (A) The adsorption of phage LP31 on different wild-type (with $O_9$ antigen) and LPS mutant (without $O_9$ antigen) strains of *S*. Enteritidis (C50041) and *S*. Typhimurium (D6). (B) Effect of added *S*. Enteritidis LPS on the ability of phage LP31 to adsorb to *S*. Enteritidis C50041. LP31 was added to *S*. Enteritidis (C50041) with either water or the same volume of LPS (100 mg/mL). (C) Transmission electron microscopy of *S*. (Continued on next page)

**FIG 4** (Continued)

Enteritidis C50041 shows adsorbed phage LP31 (the red arrows are pointing at the phages). 1: Adsorption of phage LP31 to C50041. 2: Lack of adsorption of phage LP31 to C50041$\Delta rfaL^{58-358}$. 3: Adsorption of phage LP31 to C50041 after incubation with $H_2O$. 4: Lack/decrease of adsorption of phage LP31 to C50041 after incubation with LPS. 5: Lack/decrease of adsorption of phage LP31 to C50041 after incubation with protein Lp35 [tail protein (RBP) of phage LP31] + $H_2O$. 6: Impact on adsorption of phage LP31 to C50041 when first incubated with protein Lp35 (phage RPB) + LPS (host receptor). **, $P < 0.01$; ***, $P < 0.001$.

for phage LP31 is a polysaccharide rather than a membrane protein due to the fact that the adsorption potential of phage LP31 to proteinase K-treated C50041 *S.* Enteritis was almost unchanged, while the adsorption potential to sodium periodate-treated C50041 decreased significantly (Fig. 2C) (38). The adsorption potential of LP31 to the transposon-mediated LPS synthesis mutants was also significantly reduced (Fig. 2D) ($P < 0.01$), also demonstrating that phage LP31's adsorption receptor is likely to be part of the LPS of *S.* Enteritidis.

LPS is the main component of the outer membrane of Gram-negative bacteria, consisting of O-antigen, core polysaccharide, and lipid A. The O-antigen can be classified into different serotype types (30, 39). Loss of the O-antigen results in rough LPS, and rough *Salmonella* auto-aggregate and agglutinate with the O antibody (40). Seven (*rfbF*, *rfbG*, *rfbH*, *rfbA*, *rfbB*, *rfbC*, and *rfbP*) of the 10 genes identified as being important in supporting phage LP31 adsorption are involved in O-antigen synthesis, and one (*rfaL*) is involved in connecting O-antigen to the LPS core polysaccharide. Two other genes (*rafJ* and *rfaQ*) are involved in the synthesis or assembly of LPS core polysaccharides (Fig. 2B and Table 1) (37). The *rfaL* gene of *Salmonella* encodes O-antigen ligase, and a mutant defective in *rfaL* expression may lack the ability to decorate their LPS core antigen with the O-antigen, causing LPS to transition from a smooth to a rough type (41). To verify whether the assembly of O-antigen in *S.* Enteritidis is affected by deletion of the *rfaL* gene, we constructed a *S.* Enteritidis C50041$\Delta rfaL^{58-358}$ strain, which does not aggregate in the presence of $O_9$ mAB but does aggregate in the presence of acriflavine (Fig. 3C). In addition, the C50041$\Delta rfaL^{58-358}$ strain shows obvious auto-aggregation characteristics (Fig. 3D). The rough phenotype exhibited by this strain is similar to the phenotype of $O_9$ antigen-deficient strain *S.* Enteritidis C50041$\Delta rfbG$ (42) and *S.* Typhimurium D6$\Delta rfbN^{83-188}$ (28). Therefore, the knockout of the *rfaL* gene in *S.* Enteritidis C50041 results in the loss of the $O_9$ antigen from LPS and the transition of LPS from a smooth to a rough type. The $O_9$ antigen and core polysaccharide of *Salmonella* LPS have been reported to play a role as receptors for some phages in adsorption (30). This study found that the binding ability of LP31 to the surface of C50041 was significantly reduced after incubation with smooth LPS *in vitro* (Fig. 4A), indicating that a part of LPS can bind to the RBPs of LP31 *in vitro*. LP31 was also found to lack the ability to adsorb to the $O_9$ antigen-deficient strains D6$\Delta rfbN^{83-188}$, C50041$\Delta rfbG$, and C50041$\Delta rfaL^{58-358}$, though phage LP31 did adsorb well to the surfaces of the wild-type strains (Fig. 4B and C). These data all support the concept that the adsorption process of LP31 requires the participation of $O_9$ antigen. In conclusion, the $O_9$ antigen of LPS is the receptor for phage LP31 adsorption.

RBPs of phages can specifically recognize receptors on the surface of bacteria and play an important role in their host range (43). Tail proteins of phages infecting Gram-negative bacteria often serve as RBPs to adsorb LPS (28). Bioinformatics analysis revealed that Lp35 had high homology with tail or tail-associated proteins of six different *Salmonella* phages, and its closest homolog was the tail fiber protein of phage vB_SenS_SE1 (32), indicating that Lp35 is likely to be the tail fiber protein of phage LP31 (Fig. 5C and D). However, the tail fiber protein of phage vB_SenS_SE1 has not previously been confirmed to be the RBP. We expressed and purified the Lp35 protein (Fig. 5A) and found that it could bind to the receptor on the surface of *S.* Enteritidis C50041 and reduce the adsorption of phage LP31 to C50041 (Fig. 5B and Fig. 4C), supporting our hypothesis that Lp35 can bind to the LPS on the surface of *S.* Enteritidis to prevent the adsorption of phage LP31. To verify this hypothesis, we incubated C50041 with *S.* Enteritidis purified Lp35 or with purified Lp35 and purified LPS. Following these incubations, an adsorption assay was performed with phage LP31. Purified Lp35 was capable of

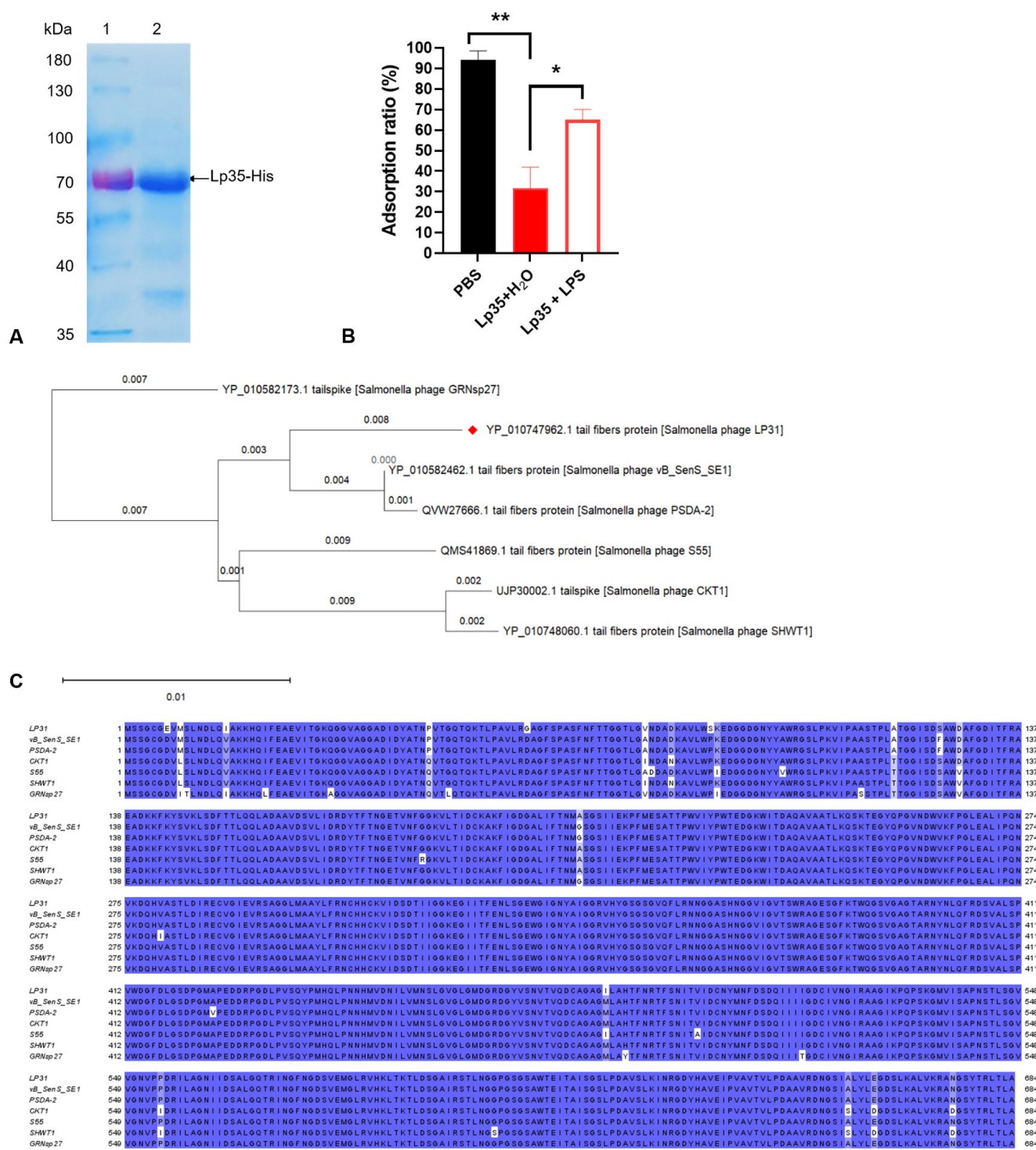

**FIG 5** Prediction and identification of the receptor-binding protein from phage LP31. (A) The recombinant protein Lp35-His was purified from *E. coli* BL21-p*lp35* and detected by Coomassie blue staining of the SDS-PAGE gel. (B) Adsorption identifies the function of the Lp35 protein during the adsorption of *S*. Enteritidis C50041 by phage LP31. *, $P < 0.05$; **, $P < 0.01$. (C) Evolutionary relationships of phage LP31 based on Lp35 phylogenetic analysis (neighbor-joining method with bootstrapping, $n = 500$). (D) Amino acid sequence alignment analysis. Identical residues are shaded in color 1, residues sharing >75% homology are shaded in color 2, and those sharing >50% homology are shaded in color 3. Color identities are given under the alignment. Numbering is based on the N-terminal methionine. The names of phages are indicated on the left.

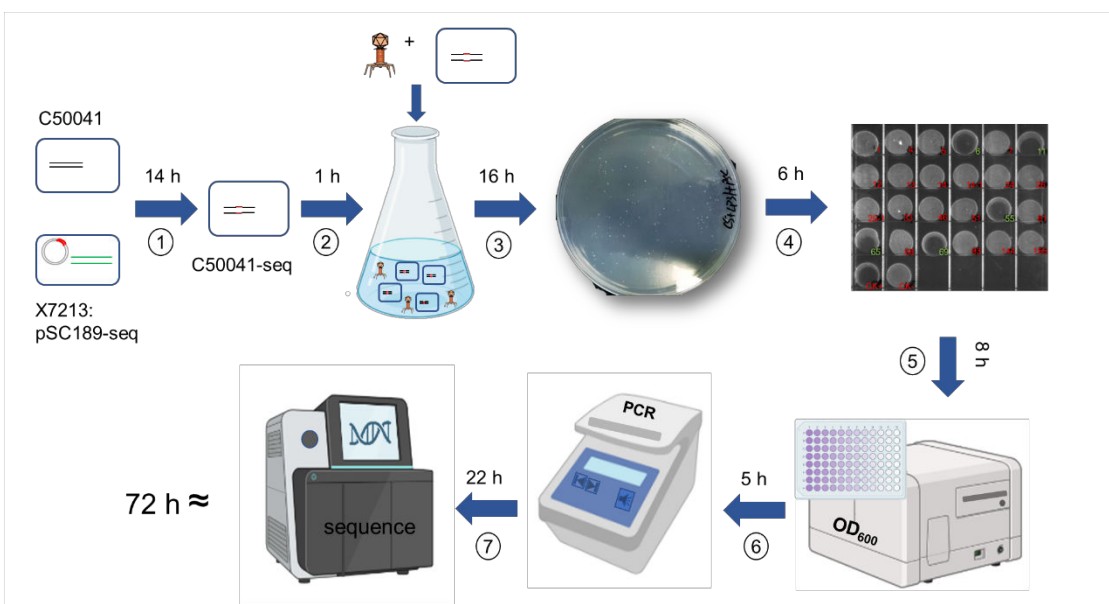

**FIG 6** Diagrammatic flow of the screening process to identify phage infection-related genes. ① Construction of a random insertion transposon mutant library. The library was constructed by the introduction of pSC189-Seq by conjugation, which carries the transposon, into the *S.* Enteritidis strain. The transposon is randomly integrated into various locations across the *Salmonella* genome. ② The transposon library was cultured in liquid media and either mock-infected or actually infected with phage LP31. After infection, the cultures were incubated for 1 further hour at 37°C to enable the phage to kill most of the phage-sensitive cells. ③ Putative phage-resistant mutants were recovered by centrifugation, and the supernatant was discarded. The bacterial pellet was washed twice with LB medium before being spread onto an LB agar plate, and the putative phage-resistant colonies were allowed to grow at 37°C for 16 h (images from Fig. 1A were reused in this figure to better illustrate the process and results of the experiment). ④ Phage-resistant phenotypes from putative phage-resistant mutants were confirmed by picking individual colonies from step 3 and growing them in 1 mL broth cultures to an O.D.$_{600}$ ≈ 0.3. Then, 20 µL of these cultures was dropped onto LB agar, allowed to dry, and exposed to phage by spotting 4 µL of an LP31 ($10^7$ PFU/mL) phage suspension on top of the bacterial spots. The putative mutants were then screened for phage LP31 resistance and sensitivity (images from Fig. 1B were reused in this figure to better illustrate the process and results of the experiment). ⑤ Resistance of the putative LP31-resistant mutants was confirmed by mixing phage LP31 at an MOI = 10 in a 96-well plate, which was then incubated at 37°C for 8 h. The growth rate for each mutant and wild type with and without LP31 was measured using a microtiter plate reader. ⑥ The transposon insertion sites in the LP31-resistant mutant were located following PCR amplification of the transposon end sequences through the sequences at the insertion positions using a primer set (Table S1). ⑦ The gene (S) interrupted by the transposon was identified following sequencing and BLAST analysis of the PCR products. If the host bacterium is *Salmonella*, the entire screening process takes about 72 h.

blocking the adsorption of phage LP31 to C50041, and Lp35 interference was relieved by allowing Lp35 to first interact with LPS before setting up the phage LP31 adsorption assay (Fig. 5B and Fig. 4C), indicating that Lp35 is able to bind to *S.* Enteritidis LPS *in vitro*. In summary, the adsorption receptor of phage LP31 is the O$_9$ antigen of *S.* Enteritidis LPS, and the RBP is tail fiber protein Lp35.

The use of a lytic phage to screen a transposon library that has survived phage infection is a robust strategy to identify mutants that have lost genes important to supporting phage infection. We narrowed down the mutants from our library that did this through the loss of a receptor by demonstrating not only that the mutants did not support phage infection but also that they no longer supported phage adsorption. We confirmed this by making a clean deletion mutant and then complementing this mutant, which showed the appropriate gain and loss of adsorption at the cell population level (adsorption assays) and at the cellular level (TEM imaging). This clean mutant analysis actually provides proof that the O-antigen is the receptor for phage LP31 and demonstrates that the strategy described here is a quick and effective method to define phage receptors. The second part of the study attempted to define what part of phage LP31 actually binds to the O-antigen. The design of these experiments was informed by bioinformatic predictions. Once the number of predictions was limited to a testable number of potential RBPs, these genes could be cloned and their products purified for use in competition assays against the known receptor. These strategies are transferable

to other phage host systems, and it is hoped that they can be used to rapidly expand our knowledge of phage host interactions and also enable more detailed studies of exactly how the RBPs interact with receptors by enabling the generation of site-directed mutants in both RBPs and receptors that can be screened rapidly for function. Together, this knowledge will help inform strategies to pick phages, build phage cocktails, and utilize the potential power phages have to control bacteria in clinical, agricultural, industrial, and other everyday settings.

## MATERIALS AND METHODS

### Bacterial strains and growth conditions

All *Salmonella* and *Escherichia coli* strains were cultured in LB (Luria-Bertani) medium at 37°C and stored in 20% glycerol at −80°C until use. Phage LP31 (GenBank accession no. OL436139) was stored in 20% glycerol at −80°C. The characteristics of the strains and plasmids used in this study are presented in Table 2. All primers used in this study are listed in Table S1. 2,6-Diaminopimelic acid (100 µg/mL), chloromycetin (Cm, 25 µg/mL), and kanamycin (Km, 50 µg/mL) were used when required. Expression of His-tagged proteins was induced by 1 mM iso-propyl β-D-1-thiogalactopyranoside (IPTG).

### Transposon mutagenesis and selection of phage-resistant mutants *S.* Enteritidis

The transposon-carrying plasmid, pSC189 from *E. coli* X7213 λpir, was transferred to *S.* Enteritidis C50041 by conjugation (28). Phage LP31 (1 mL, $10^8$ pfu/mL) was added to every 1 mL of mutant library mixture ($10^8$ CFU/mL), mixed well, and incubated at 37°C for 1 h. The infected cells were recovered by centrifugation (2,500 rcf, 5 min) and then

**TABLE 2** Strains, plasmids, and phages used in this study[a]

| Strain, plasmid, and phage | Relevant characteristics | Reference |
|---|---|---|
| *Escherichia coli* strains | | |
| X7213 λpir | Host for π requiring plasmids, conjugal donor | Laboratory collection (44) |
| X7213 λpir-pSC189 | X7213 λpir carrying pSC189, Km$^r$, Cm$^r$ | Laboratory collection (44) |
| DH5α | Δ(*lacZYA-argF*) U169 (Φ80 *LacZ* Δ*M15*) | Purchased from Takara |
| BL21 | F-, *ompT*, *hsdSB* (rB-mB-), *gal*, *dcm* | Purchased from Takara |
| BL21-p*lp35* | BL21 carrying pET28a-*lp35*, Km$^r$ | This study |
| BL21-pET28a | BL21 carrying pET28a, Km$^r$ | This study |
| *Salmonella enterica* serovar Enteritidis | | |
| C50041 | Wild type | Laboratory collection (42) |
| C50041Δ*rfaL*$^{58–358}$ | C50041 strain deleting 58–358 aa (301 aa) of RfaL protein | This study |
| C50041Δ*rfaL*$^{58–358}$-p*rfaL* | C50041Δ*rfaL*$^{58–358}$ carrying pMMB207-p*rfaL*, Cm$^r$ | This study |
| C50041Δ*rfaL*$^{58–358}$-pMMB207 | C50041Δ*rfaL*$^{58–358}$ carrying pMMB207, Cm$^r$ | This study |
| C50041Δ*rfbG* | In-frame deletion in *rfbG* | Laboratory collection (42) |
| *Salmonella enterica* serovar Typhimurium | | |
| D6 | Wild type, Carb$^r$ | Laboratory collection (28) |
| D6Δ*rfbN*$^{83–188}$ | D6 strain deleting 83–188 aa (106 aa) of RfbN protein, Carb$^r$ | Laboratory collection (28) |
| Plasmids | | |
| pSC189 | Transposon delivery vector, R6K, Km$^r$, Cm$^r$ | (45) |
| pDM4 | A suicide vector with ori R6K *sacBR*; Cm$^r$ | (46) |
| pMMB207 | RSF1010 derivative, IncQ lacI$^q$ Cm$^r$ P*tac oriT* | (47) |
| pET28a | pBR322-based expression vector utilizing T7*lac* promoter, Km$^r$ | Purchased from Novagen |
| Phage | | |
| LP31 | Siphoviridae | Laboratory collection (29) |

[a]The antibiotics are as follows: carbenicillin-resistant (Carb$^r$), chloramphenicol-resistant (Cm$^r$), and kanamycin-resistant (Km$^r$).

washed three times in 1 mL LB medium to remove residual phages. Finally, the cells were spread onto selective LB agar plates containing Km and incubated at 37°C for 16 h.

## Phage drop spot assays

The method of dropping phage suspensions onto bacterial lawns to visualize phage killing has been reported previously (28).

## Growth rate assays

The production of growth curves was performed as described previously by Bohm et al. (37) with the following modifications. Phage-resistant mutants were either uninfected or infected with phage LP31 at an MOI of 10 in a 96-well plate. The plates were incubated at 37°C/220 rpm for 8 h with optical density readings taken at 600 nm (O.D.$_{600}$) every 1 h with vigorous shaking before each read. A microplate reader (Tecan) was used to monitor the growth rates of cultures with and without phage, as described. A smaller-scale assay was also performed with only the wide-type strain, C500041, its *rfaL* deletion mutant derivative (C50041Δ*rfaL*$^{58–358}$), and the complemented strain (C50041Δ*rfaL*$^{58–358}$-p*rfaL*) incubated at 37°C/220 rpm in test tubes (5 mL LB), and the O.D.$_{600}$ was determined by spectrophotometry.

## Determining transposon insertion sites

The method refers to previous research (28). A first round of PCR amplification (primers: AB1, AB2, AB3, SP1, Table S1) was performed using genomic DNA from the phage-resistant mutants as a template; a second round of PCR (primers: ABS, SP2, Table S1) was performed using the PCR product from the first round as a template; and the second PCR product was sent to Tsingke Biotechnology Co., Ltd. for sequencing with primer pSC189-seq (Table S1). Blast analysis (https://www.ncbi.nlm.nih.gov) was performed to determine the mutant gene based on the sequencing results.

## Adsorption assays

Adsorption assays were conducted as previously described (28). Briefly, to monitor phage adsorption, phage LP31 was mixed with a fresh bacterium culture to reach an MOI of 1. After incubation at 37°C for 10 min, the phage-bacteria mixture was centrifuged at 10,000 rcf for 10 min. The free phage titer in the supernatant was determined by the double-layer agar plate method.

$$\text{Percent adsorption} = \frac{\text{pfu}^{\text{added}} - \text{pfu}^{\text{supernatant}}}{\text{pfu}^{\text{added}}}$$

## Identification of phage receptor type by proteinase K or sodium periodate treatment

To determine whether the phage receptor displayed on the host cell surface was more likely to be a protein or a carbohydrate, a total of 1.5 mL of *S.* Enteritidis C50041 culture (O.D.$_{600}$ ≈ 0.4) were treated with either 15 μL proteinase K (20 mg/mL) or 1.5 mL Ac (NaIO$_4$) [50 mM sodium acetate (pH 5.2) containing 100 mM NaIO$_4$] at 37°C for 1 h (38, 48). The treated bacteria were then washed with PBS three times and tested in an adsorption assay for their ability to support phage LP31 binding.

## Mutant strain construction

The *rfaL* gene deletion mutant strain was constructed through directed homologous recombination. The DNA fragments homologous to the upstream (primers: Up-F/R) and downstream (primers: Down-F/R) regions of the *rfaL* gene were amplified by PCR. The plasmid pDM4 was digested using the restriction endonucleases *Sac I* and *Xho I*. The amplified upstream and downstream sequences of *rfaL* were cloned into the digested

pDM4 using ClonExpress Ultra One Step Cloning Kit (Vazyme) to construct the recombinant plasmid pDM4-Δ*rfaL*. The plasmid pDM4-Δ*rfaL* was introduced into *E. coli* DH5α and X7213 λpair through CaCl$_2$-mediated transformation, in turn. Finally, the recombinant plasmid was introduced into *S*. Enteritidis C50041 by conjugation, and transconjugants were selected for the LB agar plates (15% sucrose) (the sucrose provided selective pressure for the *rfaL* gene deletion).

To complement the *rfaL* deletion, the *rfaL* gene was amplified (primers: *rfaL*-F/-His-R (Table S1), template: genomic DNA from the C50041). The plasmid pMMB207 was digested by restriction endonucleases *Xba I* and *Hind III*, and the amplified *rfaL* gene was cloned into pMMB207 using the ClonExpress Ultra One Step Cloning Kit, producing the recombinant plasmid pMMB207-*rfaL*-His. The plasmid pMMB207-*rfaL*-His was introduced into the *E. coli* DH5α and X7213 λpair by CaCl$_2$-mediated transformation. Finally, the recombinant plasmid was introduced into the deleted strain by conjugation, and transconjugants were selected for the LB agar plate containing Cm.

## Construction of vectors to produce histidine tagged recombinant proteins

To confirm the identity of the BRP of phage LP31, the *LP31GM_0000035* (*lp35*) gene was amplified by PCR (primers: *lp35*-F/-His-R (Table S1), template: genomic DNA from the phage LP31) and inserted into the pET28a vector to construct the recombinant plasmids pET28a-*lp35*-His. The plasmid pET28a-*lp35*-His was transformed into *E. coli* BL21 by CaCl$_2$-mediated transformation, and the expression of His-tagged-Lp35 was then induced using 1 mM IPTG at 19°C for 24 h. The purification and identification of the protein were performed according to the instructions of the His-tag Protein Purification Kit (Beyotime).

## Auto-aggregation assay

The auto-aggregation assay was performed based on the method previously described (42, 49). Overnight culture (30 µL) was used to inoculate 3 mL of LB medium, which was then incubated at 37°C for 16 h. The upper 100 µL was carefully removed to measure its O.D.$_{600}$, recorded as $O_1$. The remaining culture in the test tube was then vortexed to resuspend the aggregated cells, 100 µL of the suspension was removed, and its O.D.$_{600}$ was measured (recorded as $O_2$).

$$\text{Percent aggregation} = \frac{O_2 - O_1}{O_2}$$

## Agglutination assay

The agglutination assay was performed based on the method previously described (42, 49). An $O_9$ mAB (a monoclonal antibody that specifically recognizes $O_9$ antigen) (20 µL, 50 µg/mL; BioChek) or acriflavine solution (20 µL, 5 mg/mL; Hopebiol) was dropped onto a slide. A small amount of the bacterial colony was taken with an inoculation loop and mixed evenly into either the $O_9$ mAB or the acriflavine solution. The slide was gently shaken, and agglutination, if present, was observed after 1 min.

## Transmission electron microscopy

Phage LP31 and appropriate bacterial cultures were mixed at an MOI of 100 and incubated at 37°C for 10 min. The sample (20 µL) was pipetted onto a copper grid and left to sit at room temperature for 10 min, and then the excess liquid was carefully removed with filter paper. The sample was stained with a drop of 2% phosphotungstic acid for 1 min, and then the strain was removed with filter paper. The phage's adsorption on the bacteria was visualized by TEM (Hitachi, HT7800; 100 kV, ×12k–20k) (50, 51).

## Adsorption assay for phage incubated with LPS *in vitro*

Phage LP31 (500 μL of $10^4$ pfu/mL) and *S*. Enteritidis LPS (500 μL of 100 mg/mL) were mixed and incubated at 37°C for 1 h. The control group keeps other conditions the same but replaces the LPS solution with the same volume of ultra-pure water. An adsorption experiment was then performed as well as TEM imaging to observe whether LPS from *S*. Enteriditis could interfere with LP31's ability to adsorb *S*. Enteritidis C50041.

## Identification of the RBP of phage LP31

Purified protein (1 mg/mL, 500 μL) was incubated with 500 μL of *S*. Enteritidis LPS (100 mg/mL) or $H_2O$ at 37°C for 15 min. Then, *S*. Enteritidis C50041 (100 μL, $10^7$ CFU/mL) was incubated with these two solutions separately at 37°C for 15 min. Finally, the bacteria were washed three times with PBS, and the remaining 200 μL of the pellet was used for adsorption determination and TEM observation.

## Alignment and phylogenetic analysis

In order to identify the RBP of phage LP31, Jalview (version 2.11.2.6) and MEGA 11 (version 11.0.13) were used (52–54). The amino acid sequence of the tail protein Lp35 of phage LP31 was aligned and phylogenetically analyzed with proteins reported in the literature that had a Per. Identity greater than 80% with the sequences in the GenBank database (https://www.ncbi.nlm.nih.gov/genbank/). The phylogenetic analysis of the tail fiber protein of phage LP31 was performed using the neighbor-joining method with bootstrapping (*n* = 500).

## Statistical analysis

All experimental data are presented as the mean ± standard deviation of at least three independent experiments. Differences between the means of each data set were analyzed using a *t*-test with GraphPad Prism software (version 8.0.1). $P < 0.05$ was considered statistically significant.

## ACKNOWLEDGMENTS

Our work was supported by grants from the National Key Research and Development Program of China (grant 2021YFD1800403), the Jiangsu Agricultural Science and Technology Independent Innovation Funds [CX(21)1004], the Science and Technology Program of Jiangsu (BE2021331), the 111 Project (D18007), the Priority Academic Program Development of Jiangsu Higher Education Institutions (PAPD), the Fund for Excellent Doctoral Dissertations from Yangzhou University, and the Yangzhou University International Academic Exchange Funds.

All authors read and approved the submitted version of the paper. We declare no conflict of interest.

## AUTHOR AFFILIATIONS

[1]Jiangsu Key Laboratory of Zoonosis/Jiangsu Co-Innovation Center for Prevention and Control of Important Animal Infectious Diseases and Zoonoses, Yangzhou University, Yangzhou, China

[2]Key Laboratory of Prevention and Control of Biological Hazard Factors (Animal Origin) for Agrifood Safety and Quality of Ministry of Agriculture and Rural Affairs, Yangzhou University, Yangzhou, China

[3]Department of Clinical Infection, Microbiology and Immunology, Institute of Infection, Veterinary and Ecological Sciences, University of Liverpool, Liverpool, United Kingdom

## AUTHOR ORCIDs

Haojie Ge  http://orcid.org/0000-0002-1695-578X
Maozhi Hu  http://orcid.org/0000-0001-6536-193X
Heather E. Allison  http://orcid.org/0000-0003-0017-7992
Xin'an Jiao  http://orcid.org/0000-0002-2214-3358

## FUNDING

| Funder | Grant(s) | Author(s) |
|---|---|---|
| MOST \| National Key Research and Development Program of China (NKPs) | grant 2021YFD1800403 | Xiang Chen |
| Jiangsu Agricultural Science and Technology Innovation Fund (JASTIF) | CX(21)1004 | Xiang Chen |
| Science and Technology of Jiangsu | BE2021331 | Xiang Chen |
| 111 PROJCCT | D18007 | Xiang Chen |
| Priority Academic Program Development of Jiangsu Higher Education Institutions (PAPD) | | Xiang Chen |
| Fond for Excellent Doctoral Dissertations from Yangzhou University | | Haojie Ge |
| Yangzhou University International Academic Exchange Fouds | | Haojie Ge |

## AUTHOR CONTRIBUTIONS

Haojie Ge, Conceptualization, Data curation, Formal analysis, Investigation, Methodology, Resources, Software, Validation, Visualization, Writing – original draft, Writing – review and editing | Ling Ye, Resources, Writing – review and editing | Yueyi Cai, Visualization, Writing – original draft, Writing – review and editing | Huimin Guo, Resources | Dan Gu, Methodology, Supervision | Zhengzhong Xu, Methodology, Supervision | Maozhi Hu, Methodology, Supervision, Writing – review and editing | Xin'an Jiao, Funding acquisition, Methodology, Project administration, Supervision, Writing – review and editing | Xiang Chen, Funding acquisition, Methodology, Project administration, Supervision, Writing – review and editing.

## ADDITIONAL FILES

The following material is available online.

### Supplemental Material

**Fig. S1 to S3 (Spectrum02604-23-S0001.docx).** Gel images of strain construction.
**Table S1 (Spectrum02604-23-S0002.docx).** Primers designed for and used in this study.

### Open Peer Review

**PEER REVIEW HISTORY (review-history.pdf).** An accounting of the reviewer comments and feedback.

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
