## [Reviewer comments · Microbiology Spectrum]

Microbiology Spectrum

Efficient screening of adsorbed receptor for *Salmonella* phage LP31 and identification of receptor binding protein

Haojie Ge, Ling Ye, Yueyi Cai, Huimin Guo, Dan Gu, Zheng Xu, Maozhi Hu, Heather Allison, Xinan Jiao, and Xiang Chen

Corresponding Author(s): Xiang Chen, Yangzhou University

Review Timeline:

Submission Date:	June 22, 2023
Editorial Decision:	July 31, 2023
Revision Received:	August 7, 2023
Accepted:	August 14, 2023

Editor: Olaya Rendueles

Reviewer(s): Disclosure of reviewer identity is with reference to reviewer comments included in decision letter(s). The following individuals involved in review of your submission have agreed to reveal their identity: Stanley Maloy (Reviewer #2)

Transaction Report:

DOI: <https://doi.org/10.1128/spectrum.02604-23>

July 31, 2023

Prof. Xiang Chen
Yangzhou University
Yangzhou
China

Re: Spectrum02604-23 (Efficient screening of adsorbed receptor for *Salmonella* phage LP31 and identification of receptor binding protein)

Dear Prof. Xiang Chen:

Thank you for submitting your manuscript to Microbiology Spectrum. As you will see your paper is very close to acceptance. Please modify the manuscript along the lines the reviewers have recommended. As these revisions are quite minor, I expect that you should be able to turn in the revised paper in less than 30 days, if not sooner. If your manuscript was reviewed, you will find the reviewers' comments below.

When submitting the revised version of your paper, please provide (1) point-by-point responses to the issues raised by the reviewers as file type "Response to Reviewers," not in your cover letter, and (2) a PDF file that indicates the changes from the original submission (by highlighting or underlining the changes) as file type "Marked Up Manuscript - For Review Only". Please use this link to submit your revised manuscript. Detailed instructions on submitting your revised paper are below.

Link Not Available

Sincerely,

Olaya Rendueles

Reviewer comments:

Reviewer #1 (Comments for the Author):

Ge and colleagues present a novel method to elucidate the receptor binding site of phage LP31, infectious for specific strains of *Salmonella enterica*. Ge et al. constructed a random insertion transposon mutant library to identify phage resistant LPS mutants, and then created knockout strains which were subjected to agglutination, phage adsorption studies, and TEM confirm that the O9 antigen was the bacterial phage receptor site. Gu et al. also carried out further adsorption studies to identify phage protein Lp35, a tail fibre protein, involved in the binding of phage LP31 to O9. This work is timely given the rise in drug-resistant *Salmonella* spp. in many locations worldwide. Moreover, the techniques presented could be applied to many different phage-bacteria combinations to accelerate phage therapy. The manuscript is well written, the methods well described, and the work carried out to a high standard. Therefore, I only have minor comments (detailed below).

Fig 2: The yellow tones are difficult to read. Please consider replacing these with colours that provide more contrast.

Fig 4: Please consider adding descriptive text to clarify the meaning of the arrows (i.e. that they are pointing at the phages). Please also consider indicating what protein Lp35 is as the figure appears before the text describing this in the manuscript.

Line 367: Do the 'replicates' here refer to bootstraps? Please clarify this.

Reviewer #2 (Comments for the Author):

This manuscript describes a nice and thorough approach for identifying the receptor for phage, demonstrating that the Salmonella phage LP31 adsorbs to the O9 antigen in the outer membrane. A few minor comments:
It would be useful to provide references for phage resistance caused by blocking injection (line 79) because this process is less commonly recognized vs other mechanisms (Salmonella phage P22 is a good example).
The source of phage would be useful, (including vB_SenS_SE1) for others trying to replicate this work.
On line 199 a brief summary of how you "optimized" the process would be helpful for readers.
On line 201, "resistant" would be more appropriate than resistance.
On line 255 ALI mutants are genetic, substitute "deletion mutant"

Preparing Revision Guidelines

Please return the manuscript within 60 days; if you cannot complete the modification within this time period, please contact me. If you do not wish to modify the manuscript and prefer to submit it to another journal, please notify me of your decision immediately so that the manuscript may be formally withdrawn from consideration by Microbiology Spectrum.

Dear Editor

Microbiology Spectrum

I wish to submit our revised manuscript entitled “**Efficient screening of adsorbed receptor for *Salmonella* phage LP31 and identification of receptor binding protein**” for publication in *Microbiology Spectrum*.

We are grateful to the reviewers for such a rapid response to our submission, as well as their helpful comments and suggestions, which have helped us improve our manuscript. We have revised the manuscript based on all of the comments and have provided our point-by-point responses to each of their comments below. All changes are marked in yellow.

We hope that the revised manuscript is now suitable for publication in your journal.

Thank you and I look forward to hearing from you.

Sincerely

Xiang Chen,

E-mail: chenxiang@yzu.edu.cn

Reviewer #1:

Ge and colleagues present a novel method to elucidate the receptor binding site of phage LP31, infectious for specific strains of *Salmonella enterica*. Ge et al. constructed a random insertion transposon mutant library to identify phage resistant LPS mutants, and then created knockout strains which were subjected to agglutination, phage adsorption studies, and TEM confirm that the O₉ antigen was the bacterial phage receptor site. Ge et al. also carried out further adsorption studies to identify phage protein Lp35, a tail fibre protein, involved in the binding of phage LP31 to O₉. This work is timely given the rise in drug-resistant *Salmonella* spp. in many locations worldwide. Moreover, the techniques presented could be applied to

many different phage-bacteria combinations to accelerate phage therapy. The manuscript is well written, the methods well described, and the work carried out to a high standard. Therefore, I only have minor comments (detailed below).

Fig 2: The yellow tones are difficult to read. Please consider replacing these with colours that provide more contrast.

Thank you very much for your suggestion. I have replaced the yellow in Fig. 2 with golden brown colour.

Fig 4: Please consider adding descriptive text to clarify the meaning of the arrows (i.e. that they are pointing at the phages). Please also consider indicating what protein Lp35 is as the figure appears before the text describing this in the manuscript.

Thank you for your suggestion. I have added in the article that the arrow in the figure points to the phage (line 574). In addition, I have also marked the LP35 protein as the tail protein (RBP) of phage LP31 (lines 577-578).

Line 367: Do the 'replicates' here refer to bootstraps? Please clarify this.

Yes, your statement is correct. I have replaced "replicates" with "bootstrapping, n=500" (Lines 364 and 585).

Reviewer #2:

This manuscript describes a nice and thorough approach for identifying the receptor for phage, demonstrating that the Salmonella phage LP31 adsorbs to the O9 antigen in the outer membrane. A few minor comments:

It would be useful to provide references for phage resistance caused by blocking injection (line 79) because this process is less commonly recognized vs other mechanisms (*Salmonella* phage P22 is a good example).

Thank you so much for your suggestion. I have supplemented the references on blocking genomic injection according to your suggestion (line 79).

The source of phage would be useful, (including vB_SenS_SE1) for others trying to replicate this work.

In Fig. 5A, I have indicated the Genbank numbers of all phages used for sequence analysis (line 591). Additionally, I have added references to the phage vB_SenS_SE1 according to your suggestion (line 171).

On line 199 a brief summary of how you "optimized " the process would be helpful for readers.

Thank you for your suggestion. I have created Fig. 6 and provided a detailed description of the screening process. In addition, I have supplemented the optimization step (line 195) as suggested.

On line 201, "resistant" would be more appropriate than resistance.

Thank you for your suggestion. I have replaced "resistance" with "resistant" (line 197).

On line 255 All mutants are genetic, substitute "deletion mutant"

Thank you for your suggestion. I have replaced " genetic mutant " with " deletion mutant " (line 251).

August 14, 2023

Prof. Xiang Chen
Yangzhou University
Yangzhou
China

Re: Spectrum02604-23R1 (Efficient screening of adsorbed receptor for *Salmonella* phage LP31 and identification of receptor binding protein)

Dear Prof. Xiang Chen:

Your manuscript has been accepted, and I am forwarding it to the ASM Journals Department for publication. You will be notified when your proofs are ready to be viewed.

Sincerely,

Olaya Rendueles
Editor, Microbiology Spectrum
